# A Compact-Size Multiple-Band Planar Inverted L-C Implantable Antenna Used for Biomedical Applications

**DOI:** 10.3390/mi14051021

**Published:** 2023-05-10

**Authors:** Sanaa Salama, D. Zyoud, A. Abuelhaija

**Affiliations:** 1Telecommunication Engineering Department, Arab American University, Jenin P.O. Box 240, Palestine; 2Electrical Engineering Department, Applied Science Private University, Amman 11931, Jordan

**Keywords:** implantable antenna, triple-band, biocompatibility, specific absorption rate, near and far fields

## Abstract

In this paper, a compact-size multiple-band planar inverted L-C implantable antenna is proposed. The compact antenna has a size of 20 mm × 12 mm × 2.2 mm and consists of planar inverted C-shaped and L-shaped radiating patches. The designed antenna is employed on the RO3010 substrate (*ε_r_* = 10.2, *tanδ* = 0.0023, and thickness = 2 mm). An alumina layer with a thickness of 0.177 mm (*ε_r_* = 9.4 and *tanδ* = 0.006) is used as the superstrate. The designed antenna operates at triple-frequency bands with a return loss of −46 dB at 402.5 MHz, −33.55 dB at 2.45 GHz, and −41.4 dB at 2.95 GHz, and provides a size reduction of 51% compared with the conventional dual-band planar inverted F-L implant antenna designed in our previous study. In addition, the SAR values are within the safety limits with a maximum allowable input power (8.43 mW (1 g) and 47.5 mW (10 g) at 402.5 MHz; 12.85 mW (1 g) and 47.8 mW (10 g) at 2.45 GHz; and 11 mW (1 g) and 50.5 mW (10 g) at 2.95 GHz). The proposed antenna operates at low power levels and supports an energy-efficient solution. The simulated gain values are −29.7 dB, −3.1 dB, and −7.3 dB, respectively. The suggested antenna is fabricated and the return loss is measured. Our findings are then compared with the simulated results.

## 1. Introduction

Implantable antennas are the main components of implantable medical devices (IMDs); they are used for bidirectional communication in addition to external control components. At present, the design of implantable antennas is of significant interest. In the design of implantable antennas, many parameters (such as biocompatibility, miniaturization, radiation efficiency, circular polarization, and patient safety) have to be considered. Implantable antennas with minimal size and volume were proposed in [1,2,3,4,5,6,7,8,9,10,11]; however, miniaturization reduces the antenna efficiency and gain. To enhance the gain, a metamaterial technique [12,13,14] was used. In [12], a compact-size wideband implant antenna with two ring slots of the same width was presented. The proposed antenna was covered by Rogers 6010 with a high *ε_r_* of 10.2 and a thin thickness of 0.2 mm as a superstrate used for biocompatibility purposes. For gain enhancement purposes, a metamaterial superstrate with very high epsilon was printed on the existing superstrate of the implantable antenna. The results show a 3 dB gain enhancement due to the insertion of a metamaterial slab, while the resonant frequency is hardly affected by the metamaterial structure. The data rate is an issue that is critical to realizing real-time image processing for the wireless capsule endoscope (WCE). In [13], it was proven that the radiation properties of planar antennas can be enhanced by a slab of a grounded metamaterial with a high intrinsic impedance for an electric dipole source and a low intrinsic impedance for a magnetic dipole source. The optimized slab thickness is a quarter or half of a wavelength. The results in [14] show a gain enhancement of 7 dB for a patch antenna with a dielectric superstrate (Taconic Cer-10 with a dielectric permittivity of 10.2 and a thickness of 3.175 mm), which differs from the conventional patch antenna without a superstrate. In [15], zircona material with a dielectric constant of 27 was used as a superstrate for both biocompatibility issues and gain enhancement. In [16], an ultra-wideband technique was used to increase the data rate, but, on the other hand, the penetration depth of the human body is inversely proportional to the operating frequency band. To increase the transmission distance in the human body and achieve high data rate requirements, a wideband implantable antenna was designed for the human body communication band (HBC) from 10 to 50 MHz [17]. Two layers of non-equally spaced helical copper foil were used to realize the proposed antenna for a wide frequency range of 15.66 MHz at −10 dB. In [18], several techniques, such as a defected ground structure (DGS), gap coupling, and frequency-selective surfaces (FSSs), are discussed to compensate for the low-gain and narrow-bandwidth biomedical antennas. In [19], a compact-size (7 × 7 × 0.254 mm^3^) X-shaped slotted patch antenna was presented. The proposed antenna was printed on a Rogers RT5880 substrate and operated at the 915 MHz ISM band. Antenna measurements show a realized gain of −28 dB at 915 MHz. The proposed antenna in [20] provided a small-size and high-performance patch antenna for biomedical purposes. To enhance the surface current distribution and antenna efficiency, both notches and slots were combined in the antenna design.

This study is an extension of our work presented in [4,5]. In addition to a capacitive load, a planar inverted L-section is used as a parasitic element to obtain a compact-size triple-band planar inverted L-C implantable antenna. The feed point and capacitive load positions are both optimized for impedance matching purposes. The antenna design and simulation are carried out using the computer simulation technology (CST) microwave studio. This paper is structured as follows. An antenna design is presented in Section 2, while simulation and measured results are discussed in Section 3. In Section 4, the effect of skin, fat, and muscle layer thickness is given. Finally, conclusions are presented in Section 5.

## 2. Antenna Design

The proposed antenna mainly consists of two parts: the planar inverted C section (considered as an active element) and the planar inverted L section (considered as a parasitic element). The planar inverted C section has a quarter-wavelength mode (a short-circuited pin is inserted between the planar C and the ground plane). The planar inverted L section has a half-wavelength mode. In Figure 1, the C section length is 44 mm (12 + 20 + 12 mm) and the L section length is 21.5 mm (13 + 8.5 mm). The corresponding wavelengths are 176 mm and 43 mm for the planar inverted C and L sections, respectively. To calculate the related resonant frequencies, the following equation is used [21]:(1)$$f=\frac{v_{p}}{\lambda }$$
where $v_{p}$ is the phase velocity and λ is the wavelength.

The calculated resonant frequencies are 533.7 MHz for the planar C section and 2.15 GHz for the planar L section. For matching purposes at 402 MHz for the MICS band and at 2.45 GHz for the ISM band, a capacitive load of 35 pF is inserted between the L section planar and the ground plane. In addition, the positions of the short-circuited pin and the feed are both optimized for good matching at the desired frequency bands.

The antenna is printed on the RO3010 substrate (*ε_r_* = 10.2 and *tanδ* = 0.0023, with a thickness = 2 mm and a size of 20 × 12 × 2.2 mm^3^). Antenna dimensions and design specifications are summarized in Table 1.

To achieve safety conditions and prevent direct contact with the human body, the Alumina superstrate with a thickness of 0.177 mm is used (dielectric constant = 9.4 and tangent loss = 0.006). The proposed antenna is implanted in a three-layer tissue model consisting of a skin layer (with a thickness of 2 mm), a fat layer (with a thickness of 4 mm), and a muscle layer (with a thickness of 8 mm). Electrical properties summarized in Table 2 for the skin, fat, and muscle layers are used for our implantable antenna design. For each layer, the permittivity and conductivity values of 402 MHz and 2.45 GHz are defined in the dielectric dispersion fit of CST software, as shown in Table 2. The antenna is implanted in muscle at 8 mm from the skin–air interference. The whole antenna structure with a substrate and a superstrate after being implanted in a three-layer model is shown in Figure 2. The size of the three-layer model is chosen to be 70 × 60 × 14 mm^3^ to model the human chest. The antenna structure is designed and simulated using CST.

## 3. Results and Discussion

In Figure 3a,b, the return losses of the proposed implant antenna for the MICS band and the ISM band are shown, respectively. In addition, the surface current distribution and the near-electric-field distribution (402 MHz for the MICS band and 2.42 GHz for the ISM band) are shown, respectively, in Figure 4 and Figure 5. The far-field patterns for both the azimuth plane at a = 90° and the elevation plane at phi = 0° are shown in Figure 6 at 402 MHz and 2.42 GHz, respectively.

The simulated S11 characteristic for the implantable antenna structure in Figure 2 is presented in Figure 3. The results show that the proposed antenna operates at multiple frequency bands. The S11 characteristics, resonant frequency, and bandwidth for these bands are summarized in Table 3. Figure 3a shows that the designed antenna covers the MICS band from 395.57 MHz to 409.55 MHz with an S11 value of −46 dB at 402.5 MHz. For the ISM band, Figure 3b shows a wide frequency range from 2.4 GHz to 2.7 GHz with an S11 value of −33.55 dB at 2.45 GHz. In addition, a third frequency band from 2.88 GHz to 3.5 GHz is obtained with an S11 value of −41.4 dB at 2.95 GHz. For the simulated current density, Figure 4 shows that it reaches its maximum in the center at 402.5 MHz, which is equivalent to the quarter-wavelength mode. Meanwhile, at 2.45 GHz, the peak current values occur at the short edges of the design and this distribution is equivalent to the half-wavelength mode. The near electric field of the proposed antenna is also simulated, as displayed in Figure 5. The peak value of the electric field is in the center of the structure at 402.5 MHz, as shown in Figure 5a. Meanwhile, at 2.45 GHz, the peak values of the electric field occur at the short edges of the structure, as presented in Figure 5b. The antenna far-fields are also simulated using CST, as Figure 6 shows. The far-fields are approximately directed far away from the body for both MICS and ISM bands at 402.5 MHz and 2.45 GHz, respectively. The azimuth plane at (theta = 90°) is shown in Figure 6a, while Figure 6b shows the elevation plane at (phi = 0°). The simulated gain values are −29.7 dB, −3.1 dB, and −7.3 dB at 402.5 MHz, 2.45 GHz, and 2.95 GHz, respectively. The proposed antenna in Figure 1 is designed on the RO3010 substrate, as demonstrated in Figure 7. The measured and simulated results are presented in Figure 8. The measured return loss is in good agreement with the simulated results at the ISM band, while it is less matched at the MICS band. The MICS band is more sensitive to the capacitive load value than the ISM band. The capacitive load available in our lab is not exactly the same one that we used in the CST simulation, which can lead to differences between the measured and simulated results, especially at the MICS band which is more sensitive to the capacitance value.

The average specific absorption rate (SAR) is a critical parameter in the design of antennas for biomedical applications. To ensure safety conditions, the SAR value should be less than 1.6 W/Kg for the C95.1-1999 standard and less than 2 W/Kg for the C95.1-2005 standard. The simulated SAR values for 1 watt input power at 402.5 MHz, 2.45 GHz, and 2.95 GHz are summarized in Table 4 for both the 1 g and 10 g models. To satisfy safety restrictions, the input power (as a maximum) should be reduced to 8.43 mW (1 g) and 47.5 mW (10 g) at 402.5 MHz. At 2.45 GHz, the input power (as a maximum) should be reduced to 12.85 mW (1 g) and 47.8 mW (10 g). Meanwhile, at 2.95 GHz, the input power (as a maximum) should be reduced to 11 mW (1 g) and 50.5 mW (10 g). In Table 5, the maximum input power values for the 1 g and 10 g models are summarized. The simulated SAR values for the 10 g model at 402.5 MHz, 2.45 GHz, and 2.95 GHz are shown in Figure 9. In Table 6, a summary of the results for this study and other previous studies is given, and the proposed antenna covers three frequency bands with size reductions of 34% and 51% compared with the implantable antennas designed in [4,5], respectively. The proposed antenna is 528 mm^3^ in size, whereas our previous antenna designed in [4,5] were 1536 mm^3^ and 1026 mm^3^, respectively.

## 4. Parametric Study

### 4.1. The Effect of Skin Thickness

For the MICS band, as the skin thickness increases, the matching at 402.5 MHz slightly decreases, as shown in Figure 10a. Meanwhile, for the ISM band, as the skin thickness increases, the matching at 2.45 GHz decreases and a shift in the resonant frequency occurs. At 2.95 GHz, the effect of skin thickness is the same as the ISM band at 2.45 GHz, as demonstrated in Figure 10b. S11 is simulated at three different values of skin thickness (2 mm, 3 mm, and 4 mm).

### 4.2. The Effect of Fat Thickness

For the MICS band, the matching at 402.5 MHz is increased by increasing the fat thickness, as shown in Figure 11a. For the ISM band, the matching at 2.45 GHz is increased by increasing the fat thickness. The same effect of fat thickness is obtained at 2.95 GHz, as shown in Figure 11b. Three different values of fat thickness are considered (3 mm, 4 mm, and 5 mm).

### 4.3. The Effect of Muscle Thickness

By increasing the muscle thickness, both the resonant frequency and the matching for the MICS band are decreasing, as displayed in Figure 12a. Meanwhile, for ISM, by increasing the muscle thickness, the resonant frequency increases, as Figure 12b shows. The return loss is simulated at muscle thicknesses of 8 mm, 10 mm, and 12 mm. The effect of muscle thickness on S11 compared to skin and fat thickness can be seen more clearly, and this result is due to the implantation of the antenna in the muscle layer.

In addition, S11 at the ISM band is strongly affected by the three-layer thickness compared to the MICS band, and this result agrees with fact that the conductivity increases as the frequency increases [23,24].

#### The Effect of Planar Inverted L Section Length

The return loss is simulated at planar inverted L section lengths of 12 mm, 13 mm, and 14 mm. For the MICS band, it is mismatched at 402.5 MHz when the L section length becomes 14 mm, while it is well matched for L section lengths of 12 mm and 13 mm, as shown in Figure 13a. In our design, the gap between the L section and C section is optimized for good matching at 1 mm and the L-section length is 13 mm, so when the L section length is increased to 14 mm, both the L and C sections make direct contact, which leads to a mismatching at the MICS band, as seen in Figure 13a.

For the ISM band, it is well matched at 2.45 GHz and 2.95 GHz when the L section length becomes 13 mm. Meanwhile, by increasing the L section length, the resonant frequency decreases, as displayed in Figure 13b.

### 4.4. The Capacitive Load Value Effect

By increasing the capacitive load value, the matching at 402.5 MHz improves, as shown in Figure 14a. The return loss is simulated at capacitive load values of 33 pF, 35 pF, and 37 pF. Figure 14b shows that at 2.45 GHz and 2.95 GHz, the matching is very slightly affected by the three different values of the capacitive load.

## 5. Conclusions

This paper addresses the design and analysis of a compact-size multiple-band planar inverted L-C implant antenna that can be used for biomedical purposes. The proposed antenna covers a MICS band from 395.57 MHz to 409.55 MHz with an S11 value of −46 dB at 402.5 MHz, an ISM band from 2.4 GHz to 2.7 GHz with an S11 value of −33.55 dB at 2.45 GHz, and a third frequency band from 2.88 GHz to 3.5 GHz with an S11 value of −41.4 dB at 2.95 GHz. At the desired frequency bands, gain enhancement is achieved with a compact size of 20 mm × 12 mm × 2.2 mm. The antenna is simulated in a three-layer model, and the effect of skin, fat, and muscle thickness on S11 and impedance matching is presented and analyzed. The far fields are approximately directed away from the body. To ensure safety conditions, the input power as a maximum value should be reduced to 8.43 mW (1 g) and 47.5 mW (10 g) at 402.5 MHz. At 2.45 GHz, the input power as a maximum value should be reduced to 12.85 mW (1 g) and 47.8 mW (10 g), while at 2.95 GHz, the input power as a maximum value should be reduced to 11 mW (1 g) and 50.5 mW (10 g).

## Figures and Tables

**Figure 1 micromachines-14-01021-f001:**
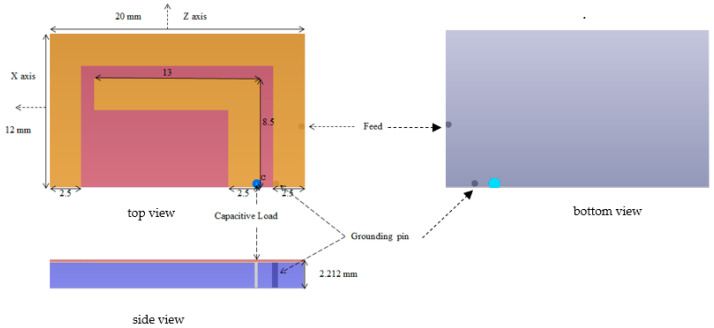
Structure of the planar inverted L-C implantable antenna (top, side, and bottom views).

**Figure 2 micromachines-14-01021-f002:**
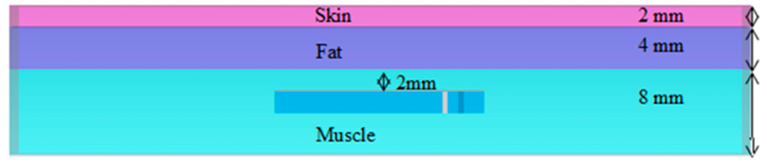
Three-layer human model.

**Figure 3 micromachines-14-01021-f003:**
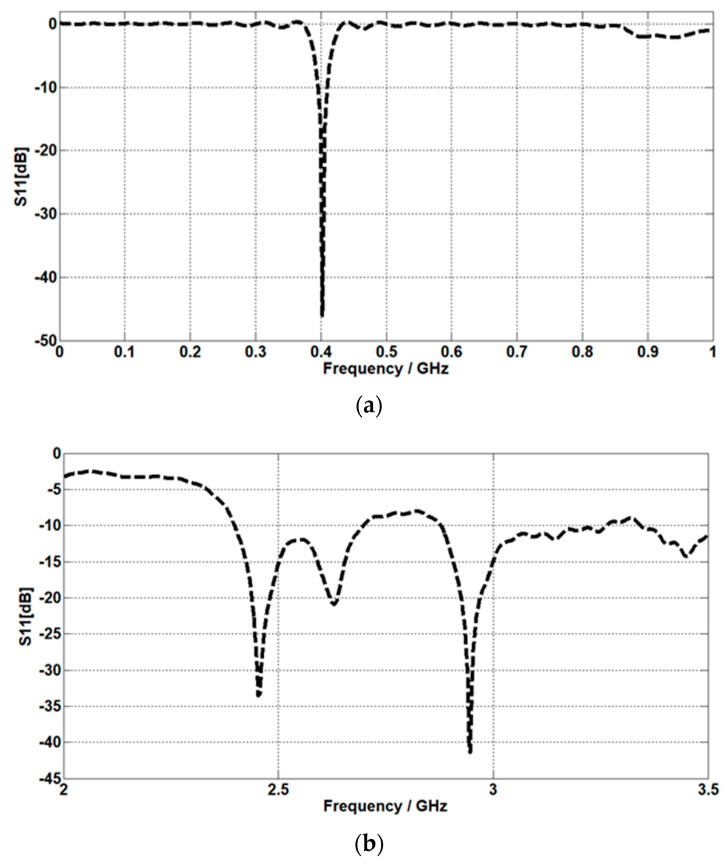
Simulated S11 values of the proposed antenna (**a**) at the MICS band and (**b**) the ISM band.

**Figure 4 micromachines-14-01021-f004:**
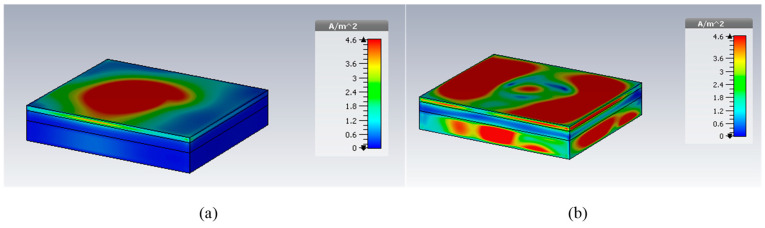
Simulated surface current densities at (**a**) 402.5 MHz and (**b**) 2.45 GHz.

**Figure 5 micromachines-14-01021-f005:**
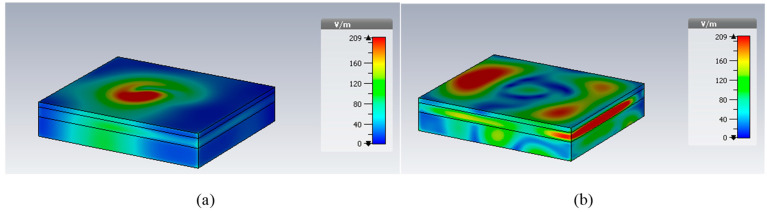
Simulated electric near-field distributions at (**a**) 402.5 MHz and (**b**) 2.45 GHz.

**Figure 6 micromachines-14-01021-f006:**
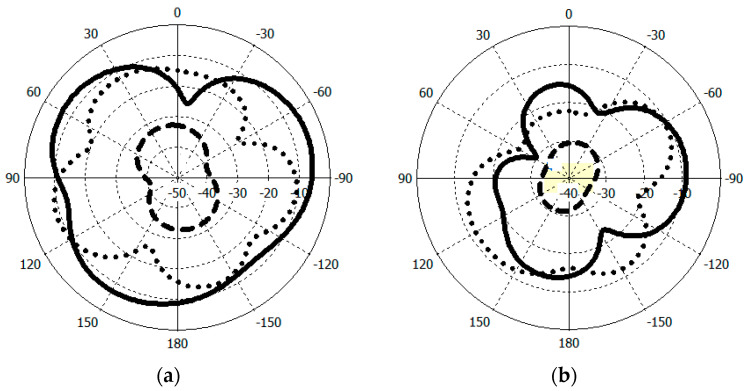
Simulated far-fields at 2.45 GHz (solid line), 402.5 MHz (dash line), and 2.95 GHz (dotted line). (**a**) H-plane; (**b**) E-plane.

**Figure 7 micromachines-14-01021-f007:**
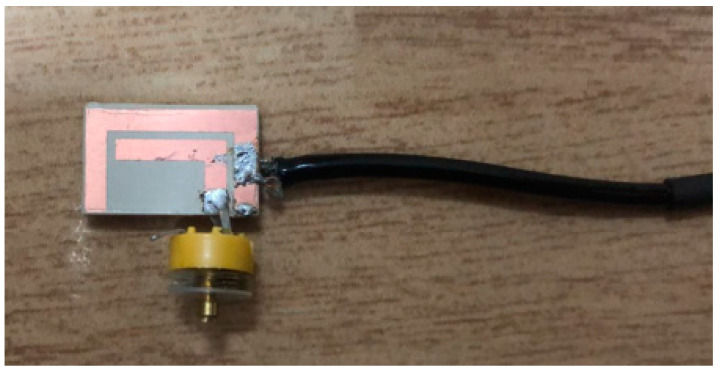
The fabricated prototype of the planar inverted L-C antenna.

**Figure 8 micromachines-14-01021-f008:**
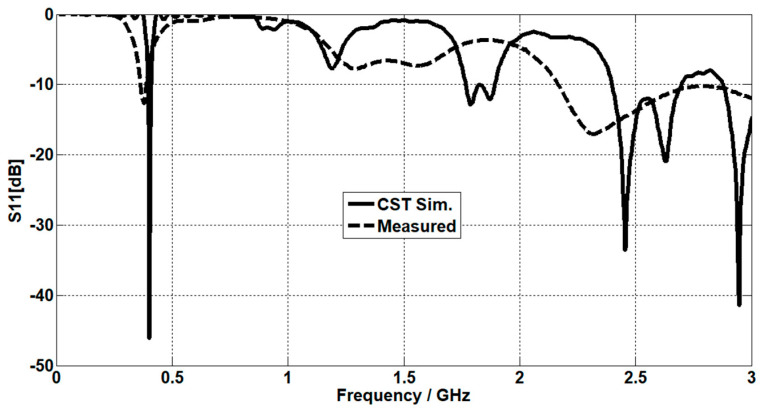
The simulation and measured results of the suggested antenna.

**Figure 9 micromachines-14-01021-f009:**
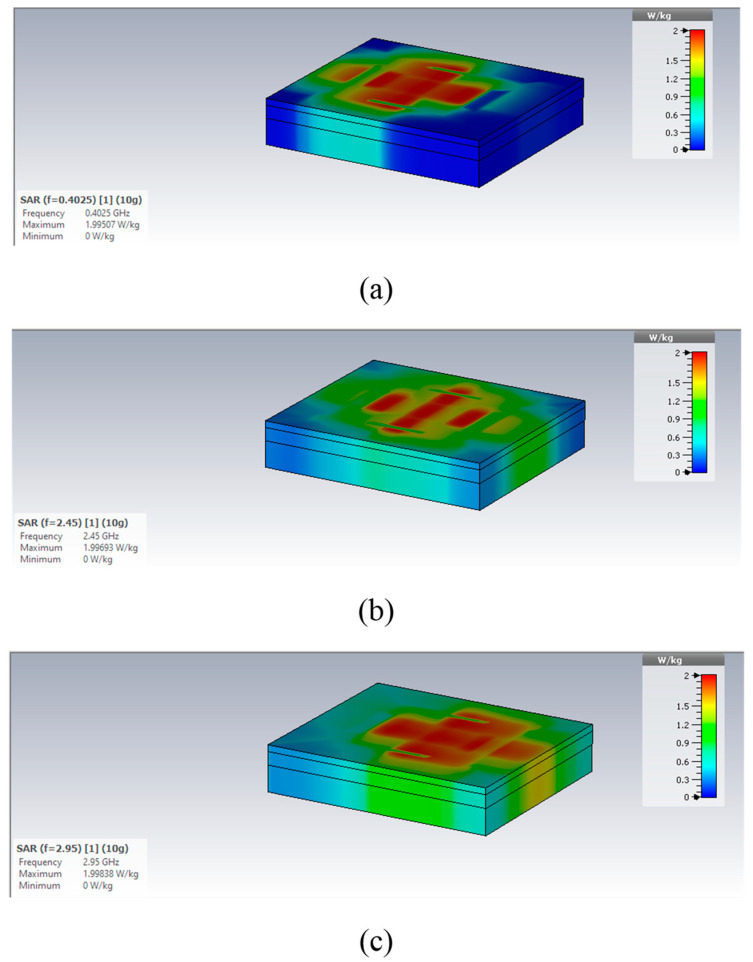
Simulated SAR values for the 10 g model at (**a**) 402.5 MHz, (**b**) 2.45 GHz, and (**c**) 2.95 GHz.

**Figure 10 micromachines-14-01021-f010:**
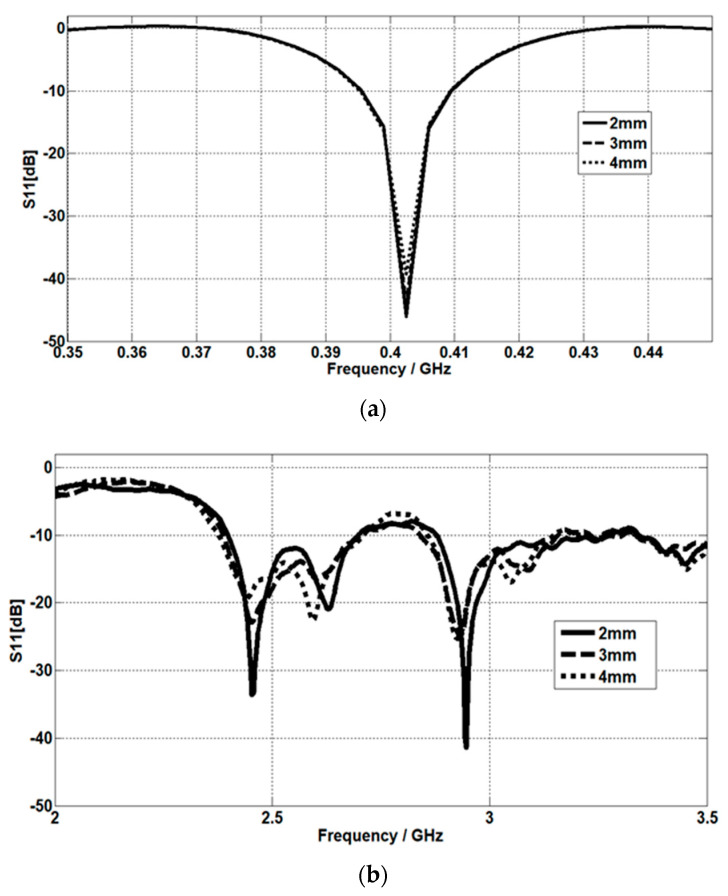
Skin thickness effect on the return loss at (**a**) the MICS band and (**b**) the ISM band.

**Figure 11 micromachines-14-01021-f011:**
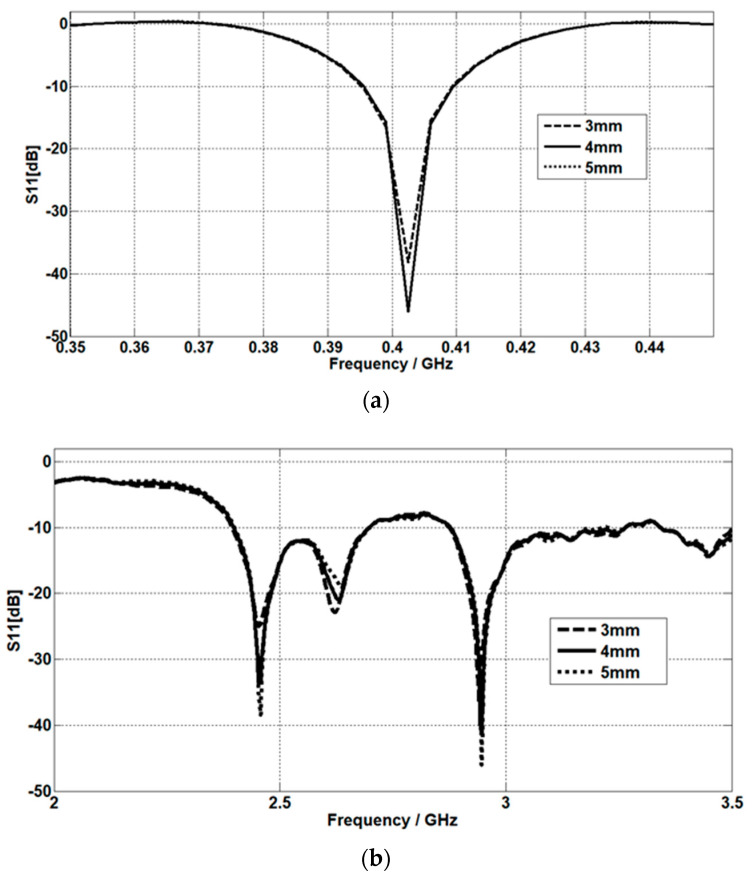
Fat thickness effect on the return loss at (**a**) the MICS band and (**b**) the ISM band.

**Figure 12 micromachines-14-01021-f012:**
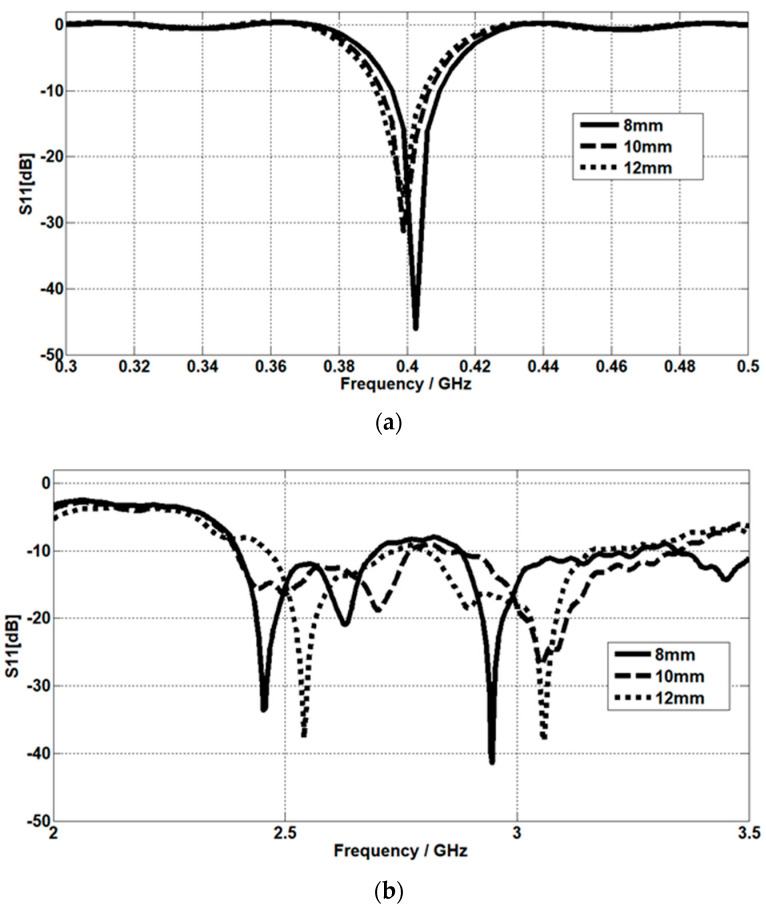
The effect of muscle thickness on the return loss at (**a**) the MICS band and (**b**) the ISM band.

**Figure 13 micromachines-14-01021-f013:**
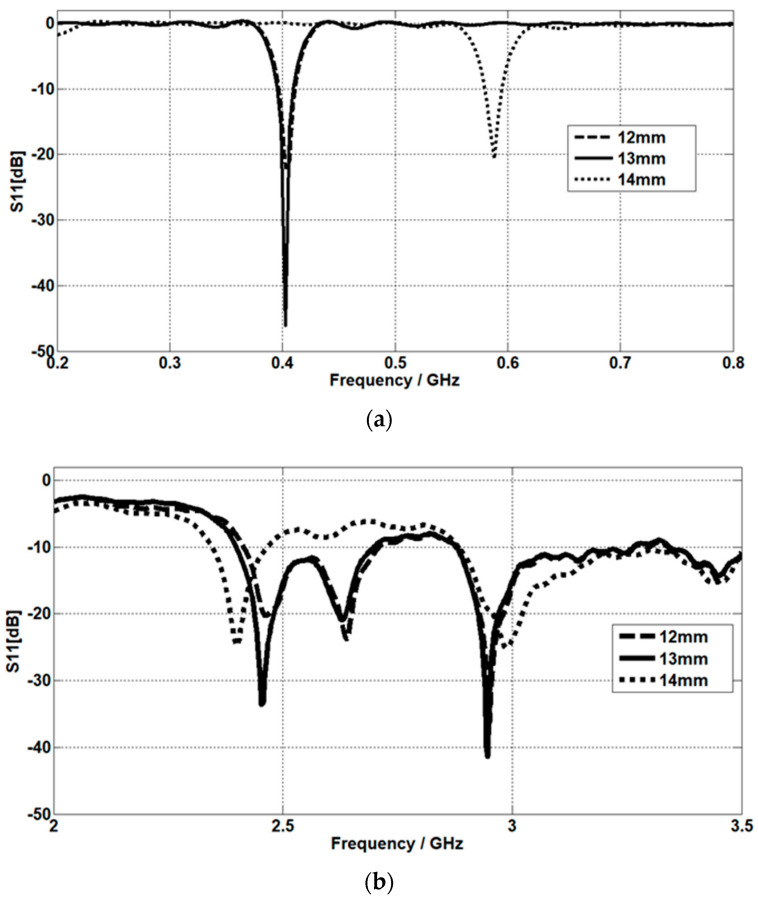
The effect of planar inverted L section length on the return loss at (**a**) the MICS band and (**b**) the ISM band.

**Figure 14 micromachines-14-01021-f014:**
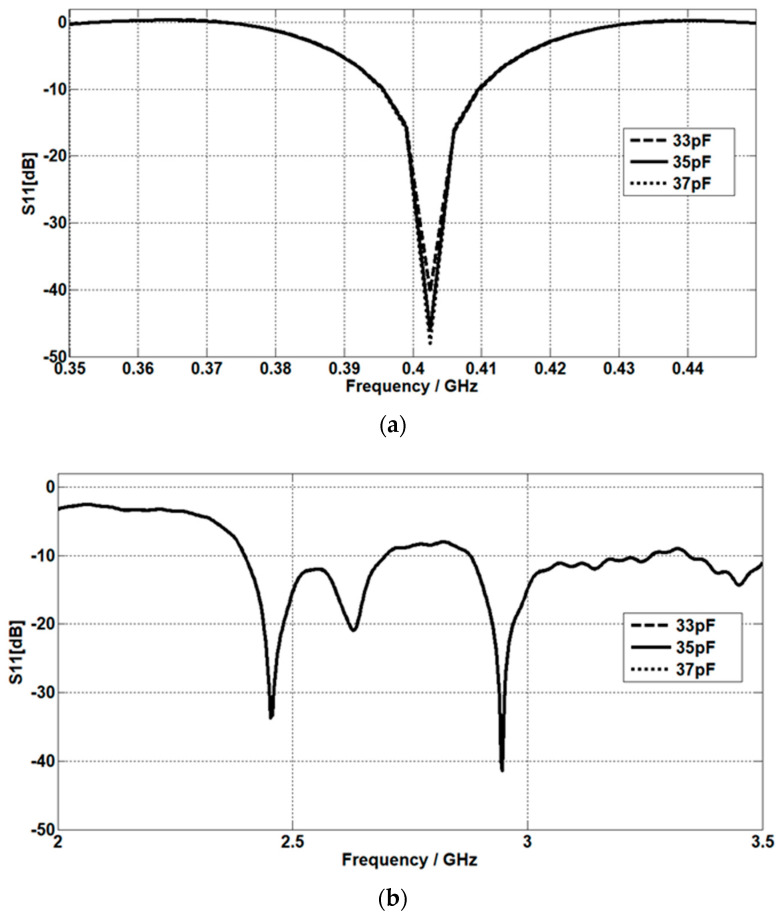
The effect of capacitive load values on the return loss at (**a**) the MICS band and (**b**) the ISM band.

**Table 1 micromachines-14-01021-t001:** Antenna dimensions and design specifications.

Planar C Element	44 mm (12 + 20 + 12 mm)
Planar L element	21.5 mm (13 + 8.5 mm)
The gap between C and L elements	1 mm
Microstrip line width	2.5 mm
Substrate	RO3010
Substrate thickness	2 mm
Superstrate	Alumina
Superstrate thickness	0.177 mm
Three-layer model	70 × 60 × 14 mm^3^
Skin thickness	2 mm
Fat thickness	4 mm
Muscle thickness	8 mm

**Table 2 micromachines-14-01021-t002:** Measured performances in different operating conditions [22].

Biological Tissues	MICs Band			ISM Band		
	$\varepsilon_{r}$	σ(S/m)	tanδ	$\varepsilon_{r}$	σ(S/m)	tanδ
Skin	46.7	0.69	0.79	38.1	2.27	0.33
Muscle	57.1	0.79	0.62	52.7	1.73	0.24
Fat	5.58	0.04	0.32	5.28	0.10	0.14

**Table 3 micromachines-14-01021-t003:** The frequency bandwidth, resonant frequency, and S11 values of the antenna structure shown in Figure 2.

Frequency Bandwidth	Resonant Frequency	Return Loss (dB)
[395.57–409.55 MHz]	402.5 MHz	−46
[2.40–2.7 GHz]	2.45 GHz	−33.55
[2.88–3.5 GHz]	2.95 GHz	−41.4

**Table 4 micromachines-14-01021-t004:** Simulated SAR values with a 1 watt input power for the 1 g and 10 g models.

Resonant Frequency	SAR (1 g Model)	SAR (10 g Model)
402.5 MHz	189.42 W/kg	42.0014 W/kg
2.45 GHz	124.246 W/kg	41.7769 W/kg
2.95 GHz	145.094 W/kg	39.572 W/kg
Standard SAR values	<1.6 W/Kg	<2 W/Kg

**Table 5 micromachines-14-01021-t005:** Calculated maximum input power for the 1 g and 10 g models.

Resonant Frequency	1 g Model	10 g Model
402.5 MHz	8.43 mW	47.5 mW
2.45 GHz	12.85 mW	47.8 mW
2.95 GHz	11 mW	50.5 mW

**Table 6 micromachines-14-01021-t006:** Comparison of the results of the suggested antenna with those of previous studies.

Reference	Frequency Band	Miniaturization Technique	Antenna Size	Substrate	Gain dBi
[2]	MICS	Folded meander line	20 mm^3^	Dermis *ε_r_* = 46.7	−23.7
[3]	ISM		816 mm^3^	FR-4	−8.5
[4]	MICS	Short-circuited pin	1536 mm^3^	Rogers-RO3010	−18
[5]	MICS and ISM	Short-circuited pin	1026 mm^3^	Rogers-RO3010	−30.14, 2.45
[19]	ISM	X-shaped slot	12.446 mm^3^	Rogers RT5880	−28
This study	MICS, ISM, and at 2.95 GHz	Short-circuited pin and capacitive load	528 mm^3^	Rogers-RO3010	−29.7, −3.1, −7.3

## Data Availability

No new data were created or analyzed in this study. Data sharing is not applicable to this study.

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
