# Peer review of "A Compact-Size Multiple-Band Planar Inverted L-C Implantable Antenna Used for Biomedical Applications"

_micromachines, 2023, doi:10.3390/mi14051021_

Round 1
Reviewer 1 Report (Previous Reviewer 1)
This paper is about the design of a compact antenna for multiband applications. The reflection coefficient confirms that the antenna provides the triple band operation. Following are the suggestions to the authors:
1. In the design section, please explain ‘44 mm (12 × 20 × 12 mm)’, it should be ‘44 mm (12 + 20 + 12 mm)’, please check and correct.
2. In the design section, the sentence ‘To calculate the related resonant frequencies, [21]:’ is incomplete.
3. The dimensions of the table should be summarized in a table.
4. The text in Table 1 seems in the form of image. Please write it in the text.
5. The title of Fig. 2 is written in sentence form. The second sentence should be included in the text not with the figure title.
6. In the title of Fig. 6, the angle information should be removed as E-plane and H-plane include this information.
7. Please explain the possible reasons for variation between the measured reflection coefficient and simulated reflection coefficient.
8. Table 5 may be moved at the end of section 4.
9. There is one line spacing between the references, please remove it.
10. Page numbers of ref [4] and [5] are missing.
11. Page numbers of ref [10] are missing.
12. Volume and page numbers of ref [15] are missing.
13. Page numbers and year of ref [19] are missing.
14. There are some grammatical and typos in the paper.
Minor editing of English language required
Author Response
Dear Reviewer;
Thank you for your comments that help to enhance my work quality.
All your comments are considered as below and highlighted as red in the main manuscript.
comment 1: is checked and corrected as 44 mm (12 + 20 + 12 mm) and 21.5mm (8.5 + 13 mm).
comment 2: the sentence is completed.
comment 3: antenna dimensions and design specifications are summarized in Table 1.
comment 4: the data in table 1 is written as text and renamed as table 2.
comment 5: the second sentence is removed.
comment 6: the angle info is removed for H-plane but for E-plane is kept the same because it is possible to be calculated for phi 0 or 90.
comment 7: the reasons for a difference between simulated and measured data are discussed within the text and highlighted as red color.
comment 8: all tables and figures will be rearranged in the final manuscript version.
comment 9: the line space between references is removed.
comment 10: page numbers of ref [4] and [5] are added.
comment 11: page numbers of ref [10] are added.
comment 12: volume and page numbers of ref [15] are added.
comment 13: Page numbers and year of ref [19] are added.
comment 14: the English language of the paper is revised.
Reviewer 2 Report (Previous Reviewer 3)
Can you add the current distribution of the antenna itself (without the human body layers) and explains the modes and different frequencies? Then you can add the current distribution with the tissues to show how it changes the modes of the antenna.
Can you add far-field measurement results of the antenna (i.e. pattern, gain)?
Can you explain why when the L section length is 14 mm, there's is a huge change to the S11 at MICS band (Fig. 13a)? It doesn't seem likely since the change is length is less than 10%.
Author Response
Dear Reviewer;
thank you very much for your comments.
all comments are considered and highlighted as gray color within the text.
regards to the L section length of 14mm:
The gap between the L section and C section is 1 mm, so when the L section length is 14 mm, then both L and C sections are in direct contact, and this leads to a mismatching at the MICs band as seen in Fig.13a.
For far field measurements, unfortunately we don't have Anechoic Antenna Test Chamber in our Lab.
Regards to current distribution it is a very good point, but for our design, the antenna structure: antenna dimensions, feed point position, capacitive load value and position are all optimized taken into account the human body layers. to get the antenna parameters in free space then this requires optimizing all antenna parameters again for matching at the required frequencies.
Reviewer 3 Report (New Reviewer)
Authors have presented a compact size multiple bands planar inverted L-C implantable antenna for biomedical applications. The work is interesting and is well supporting with simulated and measured results. Following comments will be helpful to improve the manuscript.
In abstract, it’s good to mention that the SAR values are within the safety limits with max. allowable input power (8.43 mW (1g) and 47.5 mW (10g) at 402.5 MHz; 12.85 mW (1g) and 47.8 mW (10g) at 2.45 GHz; 11 mW (1g) and 50.5 mW (10g) at 2.95 GHz) and to highlight it operated at low power and is energy efficient solution.
The text in Fig. 1 is too small and should be increased for better readability.
Table 1, appears to be an image, please replace it with a table with text.
Fig. 3, 6, 8, 10-14, are acceptable as black and white, however as its online version and for better visualization, it is suggested to use coloured versions for these figures. It will significantly increase the presentation.
Values in Fig. 9 are not readable and should be fixed.
Mentioning max input value in caption of Table 4.
Add comments to explain variations between simulated and measured results presented in Fig. 8.
Double check the manuscript for grammatical errors and typos.
Overall, the work is good.
Author Response
Dear Reviewer;
thank you very much for your comments.
all of them are considered and highlighted within the text as green color and explained as follows:
the sentence regards to SAR values is added to the abstract.
Fig.1 and Fig.9 are readjusted to be better readable.
the Tabel is rewritten as text.
the difference between simulated and measured data is explained.
standard SAR values are added to Table 4.
grammar is revised.
best regards
Round 2
Reviewer 1 Report (Previous Reviewer 1)
Authors have addressed the comments in the revised paper. Paper is acceptable. Just a minor correction: At the end of title of Fig. 6, 'phi=0 deg' should be removed. This can be corrected while submitting the final files.
Minor editing of English language required
Author Response
Dear Reviewer;
thank you very much for your efforts.
the title of Fig.6 is modified based on your comment and the language is revised.
best regards

This manuscript is a resubmission of an earlier submission. The following is a list of the peer review reports and author responses from that submission.
Round 1
Reviewer 1 Report
In this paper, authors have presented the design of a compact implantable antenna for triple band biomedical applications. The antenna parameters confirm the triple band operation. Following are the suggestions to the authors:
1. In the abstract, ‘-3.1 dB, and -7.3 dB respectively’ should be written as ‘-3.1 dB, and -7.3 dB, respectively’.
2. In the introduction, more recent literature review related to multi-band antennas for medical applications should be included.
3. Authors claim 51% reduction in size, please explain about it. Which one is the reference antenna for this comparison?
4. The equations which do not belong to the authors should be referenced.
5. Page 2: In the figure, the subfigure titles should be incorporated for top view and side view. Also, please include the bottom view of the structure.
6. How Table 1 data is achieved? If taken from a source, it must be referenced.
7. Please include the radiation patterns for all the resonant frequencies.
8. Please use a ‘,’ before the word ‘respectively throughout the paper.
9. Please check the format of the paper.
10. As mentioned above, please include more references related to biomedical antennas such as ‘X-Shaped Slotted Patch Biomedical Implantable Antenna for Wireless Communication Networks’; ‘A Wireless Data Transfer by Using a Patch Antenna for Biomedical Applications’; ‘Flexible Substrate based Printed Wearable Antennas for Wireless Body Area Networks Medical Applications’; ‘Review on Medical Implantable Antenna Technology and Imminent Research Challenges’ etc.
11. There are some grammatical and typos in the paper, please proofread the paper.
Author Response
Dear Reviewer;
First of all, I would like to thank you for your comments that help in improving my work.
Kindly note that your comments are all considered and highlighted as yellow in the revised manuscript.
best regards

Reviewer 2 Report
The paper is clearly written. It would be better to have simulations and measurements in the same plot to compare.
Author Response
Dear Reviewer;
First of all, I would like to thank you for your comments that help in improving my work.
Kindly note that our proposed antenna is fabricated and measured results are compared with simulation.
best regards

Reviewer 3 Report
The author proposed a planar inverted L-C antenna, showing operation at MICS and ISM band. However the manuscripts mainly focuses on presenting the results and does not provide sufficient analysis on how the antenna works. More importantly, no measurement results are presented to validate the design. The following are more specific comments and questions:
1. Fundamentally this antenna works as a planar inverted F antenna (PIFA) with one side open and the other side grounded and has feed in-between.
2. Is the antenna design for free space or based on the dielectric constant of the tissue layers? Are there any extra considerations taken into account when designing the antenna since it's going to be used as an implant?
3. Can the authors explain the operation modes of the antenna? Why do we need a C shape? What does the parasitic L shape do to the resonance of the antenna? How does the 35 pF capacitor change the matching?
4. How is the wide bandwidth achieved at ISM band? What parts of the antenna help increase the bandwidth?
5. What do the surface current and electric near field tell us? What can we conclude about the antenna from these plots? It might be more useful to plot the surface current of the antenna at different frequencies to give more insight about the operation of the antenna.
6. Please add experiment/measurement results.
Author Response
Dear Reviewer;
First of all, I would like to thank you for your comments that help in improving my work.
Kindly note that your comments are all considered and the proposed antenna is fabricated and measured.
- The concept is similar to a PIFA with some modifications and using parasitic element to work as a multiple band antenna.
- In the design of our antenna, the dielectric properties of the human tissues are considered at both MICs and ISM bands. In addition, for safety issues and to prevent a direct contact between the radiator and the human body, the proposed antenna is covered by a thin layer of Alumina superstrate.
- Regards to antenna modes, the C-shape element is designed to operate at the MICs band, while the parasitic L-shape element is designed to operate at the ISM band. In addition, the capacitive load value and position, and the feed point position help in bandwidth tuning and impedance matching.
- Mainly the capacitive load value and position, and the feed point position help in bandwidth tuning. In addition, the distance between the C-shape and L-shape elements is optimized because it is equivalent to a capacitive effect.
- Regards to the surface current: In Fig.4a, the surface current density is corresponding to the quarter-wavelength resonant frequency, while in Fig.4b, it is corresponding to the half-wavelength resonant frequency.
- The proposed antenna is fabricated and measured results are presented and compared with simulation results.
Best Regards
